# Correlated magnetic resonance imaging and ultramicroscopy (MR-UM) is a tool kit to assess the dynamics of glioma angiogenesis

Michael O Breckwoldt[1,2]*[†], Julia Bode[3,4][†], Felix T Kurz[1], Angelika Hoffmann[1], Katharina Ochs[2,4], Martina Ott[2,5], Katrin Deumelandt[2,5], Thomas Krüwel[6], Daniel Schwarz[1], Manuel Fischer[1], Xavier Helluy[1,7], David Milford[1], Klara Kirschbaum[1], Gergely Solecki[5], Sara Chiblak[8,9,10], Amir Abdollahi[8,9,10], Frank Winkler[5], Wolfgang Wick[5,11], Michael Platten[2,5], Sabine Heiland[1], Martin Bendszus[1][†], Björn Tews[3,4]*[†]

[1]Neuroradiology Department, University Hospital Heidelberg, Heidelberg, Germany; [2]Clinical Cooperation Unit Neuroimmunology and Brain Tumor Immunology, German Cancer Research Center, Heidelberg, Germany; [3]Schaller Research Group, University of Heidelberg and German Cancer Research Center, Heidelberg, Germany; [4]Molecular Mechanisms of Tumor Invasion, German Cancer Research Center, Heidelberg, Germany; [5]Neurology Clinic and National Center for Tumor Diseases, University Hospital Heidelberg, Heidelberg, Germany; [6]Department of Diagnostic and Interventional Radiology, University Medical Center Göttingen, Göttingen, Germany; [7]NeuroImaging Centre, Research Department of Neuroscience, Ruhr-University Bochum, Bochum, Germany; [8]German Cancer Consortium and Heidelberg Institute of Radiation Oncology, National Center for Radiation Research in Oncology, Heidelberg, Germany; [9]Heidelberg University School of Medicine, Heidelberg University, Heidelberg, Germany; [10]Translational Radiation Oncology, German Cancer Research Center, Heidelberg, Germany; [11]Clinical Cooperation Unit Neurooncology, German Cancer Consortium, German Cancer Research Center, Heidelberg, Germany

*For correspondence: michael.breckwoldt@med.uni-heidelberg.de (MOB); b.tews@dkfz-heidelberg.de (BT)

[†]These authors contributed equally to this work

Competing interests: The authors declare that no competing interests exist.

**Abstract** Neoangiogenesis is a pivotal therapeutic target in glioblastoma. Tumor monitoring requires imaging methods to assess treatment effects and disease progression. Until now mapping of the tumor vasculature has been difficult. We have developed a combined magnetic resonance and optical toolkit to study neoangiogenesis in glioma models. We use in vivo magnetic resonance imaging (MRI) and correlative ultramicroscopy (UM) of ex vivo cleared whole brains to track neovascularization. T2* imaging allows the identification of single vessels in glioma development and the quantification of neovessels over time. Pharmacological VEGF inhibition leads to partial vascular normalization with decreased vessel caliber, density, and permeability. To further resolve the tumor microvasculature, we performed correlated UM of fluorescently labeled microvessels in cleared brains. UM resolved typical features of neoangiogenesis and tumor cell invasion with a spatial resolution of ~5 µm. MR-UM can be used as a platform for three-dimensional mapping and high-resolution quantification of tumor angiogenesis.

**eLife digest** Blood vessels are the body's highways that allow blood to transport oxygen, nutrients, hormones and waste products quickly and efficiently around the body. Tumors are made up of particularly active cells and so their growth heavily depends on blood vessels. Indeed, a fundamental hallmark of tumor progression is for nearby blood vessels to form more quickly. Tumor blood vessels also differ in structure from their normal counterparts for reasons that need to be investigated in more detail.

Compounds that block the formation of blood vessels have been developed for treating highly malignant brain tumors called gliomas. However, although many of these compounds show promising effects in preclinical trials, clinical trials on humans have been less successful. Having the ability to image the blood vessels in high detail during preclinical trials would help to reveal how treatments that inhibit blood vessel formation work and how tumors might develop resistance to these drugs. However, studying tumor blood vessels remains a challenge due to technical restrictions: techniques that are able to capture how the vessels change over time are unable to show individual cells in much detail, and vice versa.

Magnetic resonance imaging is a versatile tool that can monitor how the blood vessel system of a tumor changes over time in living animals. On the other hand, ultramicroscopy is able to determine the structure of single cells of a particular type. By combining these techniques, Breckwoldt, Bode et al. have now developed a imaging platform that allows the formation of tumor blood vessels to be precisely mapped in the setting of a preclinical study. It also enables detailed investigations into how the structure of the blood vessels is altered by treatments that aim to inhibit the formation and growth of new vessels.

Using this approach on mice with gliomas, Breckwoldt, Bode et al. demonstrated that drugs that inhibit the formation of the blood vessels that supply tumors also cause the blood vessels to take on a more normal structure. Furthermore, treating the mice with a single inhibitory drug was unable to stop tumor growth, mirroring the situation in humans.

Currently, new inhibitors are being developed, offering the possibility of combined treatments that may be more effective than using a single drug on its own. The imaging platform developed by Breckwoldt, Bode et al. will allow the therapeutic effects obtained by these new treatments to be analyzed in detail during preclinical studies.

## Introduction

Gliomas are highly malignant brain tumors with poor prognosis (*Wen and Kesari, 2008*). Glioblastoma multiforme (GBM) is characterized by high cellular proliferation rates and a rapid induction of angiogenesis (*Wen and Kesari, 2008*). Angiogenesis is a hallmark of malignant tumors, and most tumors show an exponential ingrowth of neovessels upon a certain tumor size to accommodate their metabolic needs (*Carmeliet and Jain, 2011*; *Hanahan and Weinberg, 2011*). This phenomenon is called 'angiogenic switch' and is characterized by the upregulation of pro-angiogenic molecules like vascular endothelial growth factor (VEGF) or angiopoietin 2 (Ang-2) that induce fast tumor angiogenesis. The angiogenic switch is a central feature of malignant tumors and leads to a transition toward a more invasive and aggressive tumor growth (*Bruno et al., 2014*). Hence, many efforts have been made to develop antiangiogenic therapies to 'starve' tumors off their resources (*Huang et al., 2003*). So far, this approach of using anti-VEGF agents like bevacizumab has shown clinical benefit in certain tumor types (*Mittal et al., 2014*) but, despite significant improvement of progression-free survival, does not prolong overall survival of primary GBM in an unselected patient cohort (*Chinot et al., 2014*). Similarly, VEGF receptor 2 (VEGF-R2) inhibition, despite strong preclinical data (*Batchelor et al., 2007*; *Winkler et al., 2004*), has not been successful in neuro-oncology yet (*Batchelor et al., 2013*). New methods are currently developed to better identify those patients who might benefit from antiangiogenic therapies: Next to molecular approaches (*Sandmann et al., 2015*), magnetic resonance (MR) imaging could be used for patient stratification. In fact, accumulating data suggest that vascular normalization that occurs early after the initiation of therapy, might be such a marker which could aid clinical decision making (*Emblem et al., 2013*; *Lu-Emerson et al.,*

*2015*). However, improved imaging techniques that faithfully and non-invasively characterize vessel architecture and antiangiogenic treatment effects are needed to facilitate the understanding of biological actions of these therapies, and the development of clinical trials.

We employed a MR imaging approach to monitor tumor vessels at single vessel resolution. The technique is based on T2*-weighted (T2*-w) high-resolution Blood Oxygenation Level Dependent (BOLD) venography (*Park et al., 2008*) with standard gadolinium (Gd) contrast agents at ultrahigh-field strength (9.4 Tesla, isotropic 80 μm resolution). This visualizes substantially more vascular detail compared to conventional T2*-w imaging. We performed pre- and post-contrast MR scans to define angiogenesis during glioma development in two different glioma models. Also, we assess treatment effects of anti-VEGF or radiation therapy on the vascular compartment. We further mapped the tumor vascularization by correlated, dual-color ultramicroscopy (UM) of cleared, unsectioned brains (*Ertürk et al., 2012*; *Schwarz et al., 2015*). Fluorescent labeling of the microvasculature using lectins resulted in complementary 3D MR and UM data sets (dubbed 'MR-UM') of the entire tumor 'macro-' and 'microvasculature', which can be compared side-by-side. Thus, MR-UM bridges the gap between MR and optical imaging and may serve as a platform to better understand underlying mechanisms of antiangiogenic treatment and to identify novel druggable targets for preclinical therapy development.

## Methods

### Cell culture and glioma models

The mouse glioma cell line GL261 was purchased from National Cancer Institute (NCI Tumor Repository, Frederick, MD) and cultured in Dulbecco's modified Eagle's medium (DMEM) containing 10% FBS, 100 U/ml penicillin, and 100 μg/ml streptomycin (all from Sigma-Aldrich Chemie GmbH, Taufkirchen, Germany). The $1x10^5$ GL261 cells diluted in 2-μl sterile phosphate buffered saline (PBS, Sigma-Aldrich Chemie GmbH, Taufkirchen, Germany) were stereotactically implanted into the right brain hemisphere of 6- to 8-week-old female C57Bl/6J mice (n=45 mice, Charles River Laboratories, Sulzfeld, Germany) using a Hamilton syringe, driven by a fine step motor (coordinates: 2 mm lateral and 2 mm ventral of bregma; injection depth: 3 mm below the dural surface). Animals were deeply anesthetized with ketamine/xylazine and unresponsive to stimuli during the intracranial injection; GL261 cells were routinely tested for viral or mycoplasma contamination, and mouse cell identity was confirmed by the multiplex cell contamination test (Multiplexion GmbH, Heidelberg, Germany, *Schmitt and Pawlita, 2009*). The human S24 cell line was derived as a primary glioblastoma culture from a resected glioblastoma (after informed consent) and GBM typical genetic changes were confirmed by comparative genomic hybridization (*Lemke et al., 2012*; *Osswald et al., 2015*). For the S24 glioma model, $5x10^4$ S24:td-tomato cells (stably transduced by lentivirus) were transplanted orthotopically in 8- to 10-week-old male NMRI nude mice (Charles River, Sulzfeld, Germany, n=3 mice). The cells were cultivated under serum-free conditions in DMEM-F12 as sphere cultures (Thermo Fisher Scientific Inc., Waltham, MA) supplemented with 2% B-27 (Thermo Fisher Scientific Inc.), 5 μg/ml human insulin (Sigma-Aldrich Corporation, St. Louis, MO), 12.8 ng/ml heparin (Sigma-Aldrich Corporation), 0.4 ng/ml EGF (R&D Systems Inc., Minneapolis, MN) and 0.4 ng/ml FGF (Thermo Fisher Scientific Inc.). Cell lines were regularly checked for mycoplasma infections and authenticity (species control). Mycoplasma testing was done using the LookOut Mycoplasma PCR Detection Kit (Sigma-Aldrich, Germany) according to the manufacturer's instructions. All animal experiments were approved by the regional animal welfare committee (permit number: G187/10, G188/12 and G145/10, Regierungspräsidium Karlsruhe).

### MR imaging

MR imaging was performed on a 9.4 Tesla horizontal bore small animal NMR scanner (BioSpec 94/20 USR, Bruker BioSpin GmbH, Ettlingen, Germany) with a four-channel phased-array surface receiver coil. MR imaging included a standard RARE T2-w and T1-w post-Gd-contrast sequence to monitor tumor volume (T2-w parameters: 2D sequence, 78 μm in plane resolution, TE: 33 ms, TR: 2500 ms, flip angle: 90°, acquisition matrix: 200 x 150, number of averages: 2, slice thickness: 700 μm duration: 2 min 53 s; T1-w parameters: 2D sequence, 100 μm in plane resolution, TE: 6 ms, 1000 TR: ms, flip angle: 90°, acquisition matrix: 256 x 256, number of averages: 2, slice thickness: 500 μm,

duration: 5 min). To assess the tumor vasculature, we used a T2*-weighted gradient echo sequence (*Park et al., 2008*) and acquired pre- and post-contrast scans (3D sequence, 80 μm isotropic resolution, TE: 18 ms; TR: 50 ms; flip angle: 12°; number of averages: 1, acquisition matrix: 400 x 188 x 100, duration: 15 min 40 s). Pre-contrast images were used to differentiate susceptibility artifacts caused, for example, by tumor microbleedings from vessel signals that were only detectable after contrast administration. Dynamic contrast-enhanced (DCE) imaging (TE: 1.8 ms; TR: 16 ms; flip angle: 10°; slice thickness: 700 μm, acquisition matrix: 66 x 128, 3 slices acquired, number of averages: 1, 300 repetitions; 700 μm in plane resolution; duration: 10 min, time resolution 2 s) was used to assess vascular permeability ($K_{trans}$). 0.2 mmol/kg Gadodiamide (Omniscan, Nycomed, Ismaningen, Germany) was administered by tail vein injection for DCE and post-contrast scans. In five animals, 50 μl Gadodiamide was administered by intraperitoneal injection (ip), which had been determined before to match an iv dose of 0.2 mmol/kg. To assess the validity of the T2*-w sequence with an iron-based contrast agent, we performed blood pool imaging pre- and post-iv injections of crossed linked iron oxide nanoparticles (USPIO, CLIO-FITC, 15 mg/kg, particle size: 31 nm, kind gift by R. Weissleder, MGH, Boston). MR imaging was started 7 to 14 days post GL261 tumor cell implantation and repeated weekly up to 5 weeks after tumor cell implantation. For MR imaging, animals were anesthetized with 3% isoflurane. Anesthesia was maintained with 1–2% isoflurane. Animals were kept on a heating pad to keep the body temperature constant. Animal respiration was monitored externally during imaging with a breathing surface pad controlled by an in-house developed LabView program (National Instruments Corporation).

## Vascular permeability imaging

DCE imaging was post-processed with OLEA software (OLEA Medical, La Ciotat, France). Pseudo-color images represent vascular permeability maps ($K_{trans}$) that were calculated using the Tofts and Kermode model (*Tofts and Kermode, 1991*). Quantification of DCE time series was performed in FIJI by ROI analysis of the main tumor slide and encompassing the entire tumor area. A mirror ROI was set in the contralateral hemisphere. Intensity values were measured and processed in Microsoft Excel (Microsoft). Signal ratios (SR) were calculated over time (SR=$signal_{tumor}$ / $signal_{contralateral\ site}$) and displayed as blood–brain barrier disruption (BBB-D, arbitrary units, a.u.).

## Antiangiogenic therapy

GL261 glioma-bearing mice were treated with the murine-chimeric anti-VEGF antibody which is based on the humanized anti-VEGF antibody B20-4.1 (*Srivastava et al., 2014*) or murine IgG control antibody (Roche, pRED Innovation Center Penzberg) at a dose of 10 mg/kg every 3 days by ip injection (four mice per group). Treatment was initiated 2 weeks after tumor cell implantation when a solid tumor was present on MRI. Animals were treated for 1 week (days 0, 3, 6) and MRI was performed on day 7 (3 weeks post inoculation) to quantify vascularization parameters. After the MR investigation, animals were injected with lectin-FITC or lectin-texas red and sacrificed for UM correlation.

## Radiotherapy

Photon irradiation was delivered by XRAD320 X-ray device (Precision X-Ray, CT, USA). Animals were anesthetized by ketamine/xylazine during the procedure. Whole-brain irradiation was performed with a lateral photon beam at a dose of 2 grays per day. Irradiations were performed for four consecutive days starting 2 weeks after GL261 tumor implantation. The body was protected by lead shield to avoid radiation exposure to organs other than the brain. MR measurements were performed before (week 2) and after the completion of irradiation (week 3). Mice without irradiation served as controls (n=3 mice per group).

## Computed tomography

To quantify the apparent volume loss of brain tissue induced by the clearing procedure, we performed computed tomography (CT) scans before and after clearing. Two 360° scans were obtained on a small animal CT (Quantum FX, Perkin Elmer, Waltham, MA) using the following parameter: 90 kV, 200 μA, 24 mm field of view, 40 μm isotropic resolution. Image reconstruction was performed using a standard filtered backprojection algorithm implemented in the vendor's software. The

maximum brain dimensions were determined in transversal slices and compared before and after clearing. We found that the brain volumes shrinks by ~40% but that the ratio of maximum length/width stays constant before and after clearing (length and width pre clearing: $1.33 \pm 0.54$ cm x $1.05 \pm 0.38$ cm; post clearing: $0.79 \pm 0.24$ cm x $0.62 \pm 0.17$ cm; ratio: $1.25 \pm 0.01$ and $1.28 \pm 0.03$, $p>0.05$ for ratio comparison).

## Image processing and analysis of MR data

The vascularized area was quantified in Osirix software (V.4.12, Pixmeo, Geneva) by manually selecting region of interests (ROIs) around tubular vessel-like structures on post-contrast T2*-w images. Care was taken to exclude areas that were hypointense on pre-contrast images and most likely represent microbleedings or calcifications. For histogram analysis, the tumor region and an outside region in the frontal white matter were manually segmented. Intensity values for histogram analysis were read out in Matlab (Release 2015a, The MathWorks, Inc., Natick, MA). Images were normalized using the following formula: (mean voxel intensity$_{tumor}$ – intensity$_{outside}$)/standard deviation (SD).

## Labeling of the microvasculature and assessment of BBB-D

For correlative optical microscopy of the microvasculature, animals were injected with fluorescent lectins that bind to N-acetyl-β-D-glucosamine oligomers of endothelial cells (*Wälchli et al., 2015*). Injection of isolectin-FITC (12 mg/kg, Sigma) or lectin-texas red from *Lycopersicon esculentum* (12 mg/kg; Vector laboratories) was performed via the tail vein in 100 µl PBS after MRI. Animals were sacrificed 5 min after lectin injection by a ketamine/xylazine overdose and transcardially perfused with 5 ml PBS followed by 10 ml 4% paraformaldehyde (Histofix, Carl Roth GmbH, Karlsruhe). For the assessment of BBB-D, animals (n=2) were iv injected with 150 µl of 2% Evans blue (EB, Sigma) diluted in PBS. Animals were sacrificed 10 min after EB injection and cleared using the FluoClear-BABB protocol (see below). Caution must be taken when using EB in UM as prolonged animal perfusion and clearing can result in a drop of fluorescent signal due to a possible washout of the hydrophilic compound.

## Fixation and clearing of mouse brains

Brains were fixed after perfusion with 4 % buffered formalin for at least 24 hr at 4°C in the dark. For UM-analysis, whole brains were optically cleared using organic solvents. Clearing was performed according to the 3DISCO protocol (*Ertürk et al., 2011*; *2012*). Brains were transferred into glass vials for tetrahydrofuran (THF; Sigma Aldrich) dehydration. Two milliliter of 50 % THF were gently added using a pipette. Vials were placed into black 50-ml Falcon tubes and then mounted onto an overhead turning wheel (program C3, 15 rpm, Neolab, intelli-mixer). Clearing was performed at room temperature. After 12 hr, the 50 % THF solution was exchanged by a 70 % THF solution. Vials were again put into black Falcon tubes and mounted onto the turning wheel for another 12 hr. The procedure was repeated with 80% and 100% THF solutions, respectively. Samples were incubated for 12 hr in 100 % THF for three times. The dehydrated brains were placed in benzyl ether for 48 hr in order to clear the samples (98 % dibenzyl ether, DBE, Sigma-Aldrich, Steinheim, Germany). To avoid degradation of the fluorescent signal, samples were kept in the dark and imaged immediately after the clearing procedure. S24 tumor-bearing brains were cleared with the recently published protocol FluoClearBABB (*Schwarz et al., 2015*). This protocol is based on benzyl alcohol/benzyl benzoate clearing in combination with a basic pH, which is maintained throughout the clearing procedure. The protocol is especially suited for effective clearing of aged mouse brains. Mice were perfused with lectin-FITC as described. After dissection, brains were kept in PBS at 4°C. For the dehydration of brains, analytical grade alcohol (t-butanol, Sigma) was diluted with double-distilled water. Brains were dehydrated using t-butanols ranging from 30 to 100 %. The clearing solution BABB was prepared by mixing benzyl alcohol (Merck, analytical grade) and benzyl benzoate (Sigma, 'purissimum p.A.' grade) in a 1: 2 volume ratio. The pH levels of dehydration and clearing solutions were adjusted using an InLab Science electrode suited for organic solvents (Mettler-Toledo). pH levels were adjusted with triethylamine (Sigma-Aldrich).

## Acquisition of ultramicroscopy data sets

The cleared brains were scanned with a light sheet microscope (LaVision BioTec GmbH, Bielefeld, Germany). We used 0.63x, 1.0x, and 2.0x with a 2x objective lens and a white light laser (SuperK EXTREME 80 mHz VIS; NKT Photonics, Cologne, Germany) with a wavelength spectrum ranging from 400 to 2400 nm (pixel size for 0.63x: 5.16 µm; for 1.0x: 3.25 µm and for 2.0x: 1.62 µm). For the detection of blood vessels, the following filters were used: lectin-FITC, excitation 470 / 40 nm; emission 525 / 50 nm; lectin-texas red excitation 545 / 25; emission 585 / 40.

Z-stacks with 5 µm step size and a total range of up to 1500 to 2000 µm for the transversal measurement of the whole brain were acquired. Measurements with exposure times of 300 ms per slice resulted in a total acquisition time of ~10 min per brain sample and magnification. Images were exported as tagged image file (tif) and further post-processed in the ImageJ package FIJI, version 1.49 (http://fiji.sc/Fiji). For the generation of UM movies, the 'Running Z projector' plugin (FIJI) was used.

## Quantification of ultramicroscopy data

Representative single slices from light sheet microscopy data sets were scaled in FIJI to a resolution of 0.5 x 0.5 µm$^2$. ROIs were manually chosen inside the tumor region and in a region not affected by tumor or vessel alterations ('outside'). Vessels were identified with the FIJI plugin 'tubeness' that finds linear structures in an image (*Sato et al., 1998*). The resulting images were binarized individually to ensure maximal congruence with the respective vessels on the original image slice. For each binarized ROI, vessel objects were identified and skeletonized using Matlabs built-in functions 'bwmorph' and 'regionprops'. After subtracting vessel branch points of the skeletonized image, the length of each vessel segment was determined. To obtain the vessel diameter and to avoid very small vessel cross-sections with minimal length, only those segments were considered whose lengths were longer than the difference of mean and standard error of the segment length distribution. For each pixel $i_S$ of each segment $S$, the minimum distance to a non-vessel pixel was determined as the pixel-specific segment radius $r_{is}$. Subsequently, we determined average segment radius $r_S = \frac{\sum_{i_S \in S} r_{i_S}}{\sum_{(i_S \in S)}}$ and overall mean segment radius $\tilde{r} = \sum_S \overline{r_S} / \sum_S$. The tortuosity of the longest 10% of the segments was obtained by dividing segment length by the Euclidean distance between its endpoints. The quantification of the vasculature in the 3D stack was performed in Amira (FEI, Hilsboro). The tumor and an 'outside region' were segmented in UM data sets (100–200 images per stack). Segmentation of the vasculature was performed semi-automatically based on pixel intensity. Thresholded images were median filtered, skeletonized and vessel length and radius were quantified.

## Immunohistochemistry

To correlate MRI findings to histology 10 µm coronal cryostat sections were cut. Hematoxylin & eosin staining was performed for anatomical assessment. Lectin-FITC or lectin-texas red was injected iv in healthy animals (n=2) and dye labeling of endothelial cells was assessed on cryosections. Immunohistochemistry for endothelium (CD31 antibody, BD Bioscience) was performed using standard immunohistochemistry protocols and analyzed on a confocal laser-scanning microscope (Olympus FV1000). CD31 staining in lectin-FITC injected mice was performed in three GL261 tumor-bearing animals.

## Statistical analysis

Data is shown as mean ± SEM. Statistical analyses were performed in PRISM (GraphPad). Two-tailed student's t-tests were used to compare two groups. One-way ANOVA with Bonferroni's post hoc testing was used for multiple comparisons. p-Values < 0.05 were considered significant. *denotes $p<0.05$; **$p<0.01$; ***$p<0.001$.

## Results

### Tumor vascularization can be monitored by contrast-enhanced T2*-weighted MRI

Glioma-bearing mice were monitored weekly with high-field MRI (9.4 T) starting 1 week after tumor cell implantation. Tumors could be clearly visualized by Gd-enhanced T1-weighted imaging (*Figure 1a*). Tumors grew rapidly and untreated mice got symptomatic and were sacrificed 4–5 weeks post implantation when they had developed large intracranial tumors (*Figure 1a*). To monitor vascularization both in the healthy brain and under pathological conditions, we employed a gradient echo, T2*-w sequence with high spatial resolution (80 μm isotropic resolution). This sequence allows the visualization of venules due to the BOLD effect of deoxygenized blood and the resulting susceptibility signals before contrast administration (*Figure 1b*). After administration of a clinically applied Gd-contrast agent, arterioles and venules show strong tubular vascular susceptibility signals that allow the assessment of the vascularization status (*Figure 1b*). Also, an iron-based experimental contrast agent (ultrasmall iron oxide nanoparticle, USPIO) resulted in a good delineation of the vasculature (*Figure 1—figure supplement 1*). In the healthy brain, for example, basal ganglia and cortical penetrating vessels can be visualized (*Figure 1b* and *Figure 1—figure supplement 1*). Under pathological conditions arterioles and venules within the tumor bed could be identified starting 2 weeks post-tumor implantation (*Figure 1c*, and *Figure 1—figure supplement 2*). First detectable vessels run both centrally and in the periphery of the developing glioma and could only be visualized after the administration of contrast material as tubular hypointense structures (*Figure 1c*).

### Tumor vessel formation occurs within a short timespan

Within a week from the detection of first neovessels massive additional hypointense vessel-like structures occurred in the entire tumor bed as visualized on T2*-w images (*Figure 1d* and *Videos 1,2*). At this 'late' stage, single vessels were hard to differentiate within the tumor core because vascular density became too high for single vessel differentiation. Next to vascular susceptibility signals, also some dot-like susceptibility signals were found before contrast administration, most likely representing tumor microbleedings (*Figure 1d*). Kinetic studies showed that the Gd contrast agent is cleared from the circulation within ~2 hr after iv injection and vascular signals return to baseline within this time period (*Figure 1—figure supplement 3*). Quantification of the tumor size and vascularized area showed an exponential increase of tumor vessels over the time of investigation 1–3 weeks post-implantation (vascularized area week 1: 0.2 mm$^2$, week 2: 0.9 mm$^2$, week 3: 3.5 mm$^2$, p<0.01; *Figure 1e,f*). Histogram analysis of MR images showed a strong signal drop in the tumor area 3 weeks post-tumor cell implantation which was present only after contrast administration (*Figure 1 g,h*). Interestingly, MRI was sensitive enough to detect single tumor vessels starting 2 weeks post implantation (*Figure 1i*). The resolution of MRI is, however, intrinsically limited to the μm range which is insufficient to resolve tumor vessels below ~50 μm in size. To validate and correlate our MR results to the cellular level, we performed UM in combination with tissue clearing.

### Tumor vessel architecture can be visualized by light sheet microscopy

We adapted recently published light sheet microscopy protocols of cleared whole-brain specimen (*Ertürk et al., 2012*; *Schwarz et al., 2015*). Similar to MRI, UM results in 3D datasets but possess a ~10-fold higher resolution (*Keller and Dodt, 2012*). For UM, the microvasculature was assessed by lectin-FITC or lectin-texas red. These intravital dyes bind to endothelial glycoproteins in healthy tissue and tumor endothelium (*Figure 2b* and *Figure 2—figure supplement 1*, *Wälchli et al., 2015*). After MR imaging, animals were injected intravenously with lectin and sacrificed after 5 min of dye circulation. Subsequently, brains were explanted, cleared and vascularization was assessed by UM. The pathological neovascularization of the xenografted glioma was clearly visible: Single vessels in the tumor bed could be well delineated on correlated MRI and UM datasets (MR-UM) 2 weeks post-tumor implantation (*Figure 2c* and *Videos 3, 4*). Vessel density massively increased and large pathological vessels occurred 3 weeks post-tumor implantation (*Figure 2d*). In some larger tumors parts of the tumor core became necrotic and no vessels were present in these areas (*Figure 2d*). The MR-UM approach can also be employed for additional targets: for example,conventional Gd-enhanced

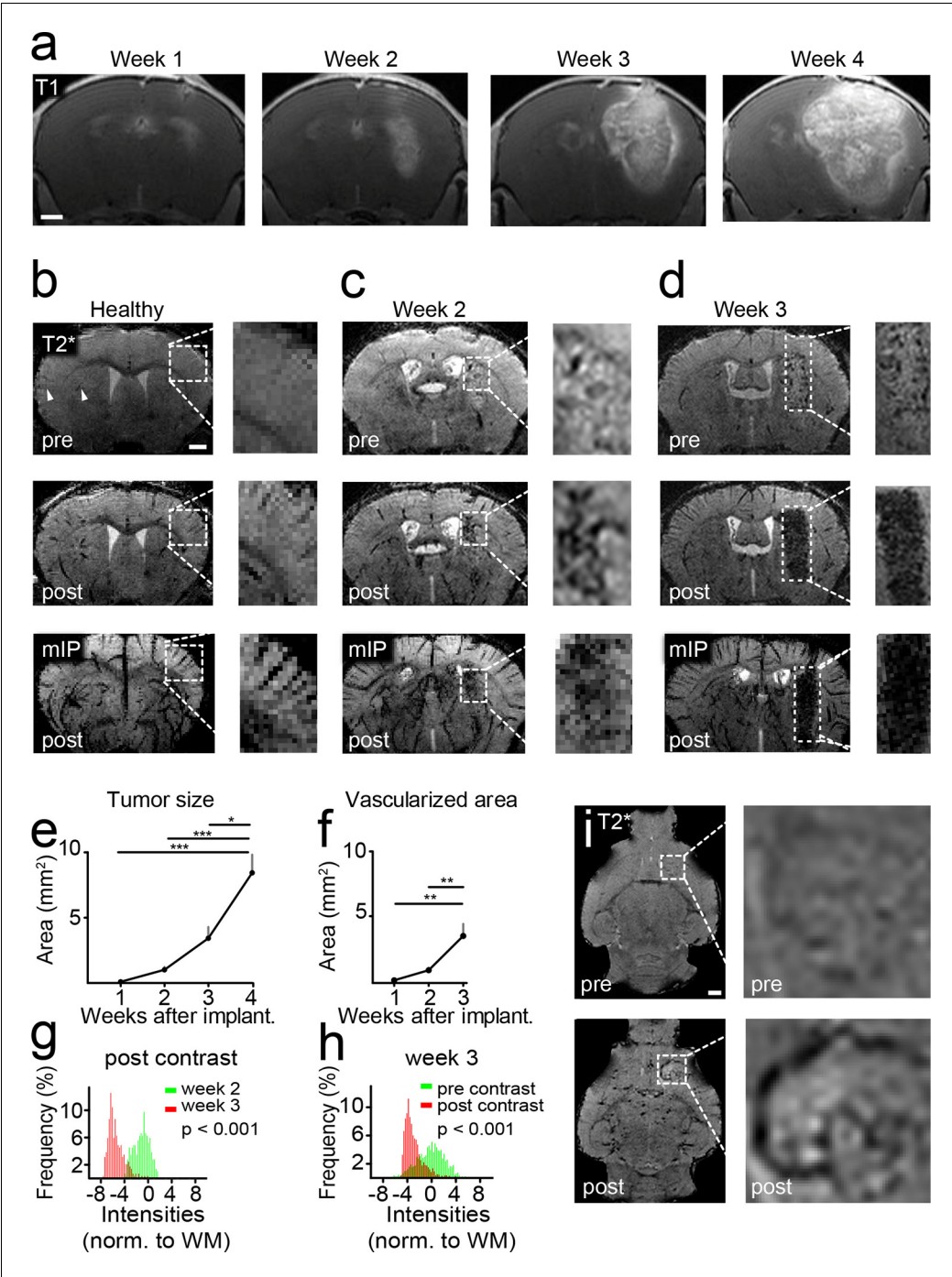

**Figure 1.** Imaging tumor vessel development with T2*-w sequences. Time course of tumor development on T1-w post Gd-contrast images (**a**). T2*-w images (80 μm resolution). Hypointense tubular structures, most likely venules (arrowheads), are visible due to the BOLD effect on pre-contrast images (upper row). Post-contrast administration arterioles and venules can be visualized (middle row). A minimum intensity projection (mIP) is shown in the bottom row (**b**). T2*-w images two and 3 weeks post-tumor implantation (**c,d**). Quantification of tumor sizes on T1-w post-contrast images (**e**). Quantification of the vascularized area on T2* images (**f**). Histogram analysis of the tumor region on post-contrast images 2 (green distribution) and 3 weeks (red) after tumor implantation. A significant signal drop (red distribution) within the tumor relative to the healthy white matter (WM) occurs within 1 week (p<0.001, **g**). This signal drop is only visible post-contrast administration (red distribution, p<0.001, **h**). Single plane T2*-w images pre- and post-contrast 2 weeks post-tumor implantation. The tubular vessel is only visible after contrast injection (**i**). Note the tortuous appearance and the multiple branches of the vessel. Scale bars are 1 mm.

The following figure supplements are available for figure 1:

*Figure 1 continued on next page*

**Figure supplement 1.** Monitoring vascularization with USPIOs in healthy mice.
DOI: 10.7554/eLife.11712.004
**Figure supplement 2.** Imaging tumor vasculature with USPIOs.
DOI: 10.7554/eLife.11712.005
**Figure supplement 3.** Gd contrast kinetics in T2*-w time series Sequential.
DOI: 10.7554/eLife.11712.006

MRI can easily be correlated to optical Evans blue extravasation studies to assess BBB-D and vascular permeability within the tumor ($R^2$: 0.87, p<0.001; *Figure 2—figure supplement 2a–c*).

## S24 tumors exhibit markedly different growth and angiogenesis patterns

The comparison of growth and angiogenesis patterns of primary, human S24 tumors with syngeneic mouse GL261 tumors by MR-UM revealed striking differences: S24 tumors grew markedly slower and highly invasive compared to GL261 tumors. Individual S24 tumor cell clusters were found outside of the tumor bulk even in the contralateral hemisphere. BBB-D, an early feature of GL261 tumors, developed in S24 tumors only at 10 weeks post-tumor implantation (*Figure 2—figure supplement 3*). At this time point, S24 tumors started to show clear signs of increased vascularity at the tumor stroma border on T2* post-contrast images and correlated UM (*Figure 2e*, *Figure 2—figure supplement 3* and *Video 5*). UM also confirmed tumor cell infiltration into the contralateral hemisphere, which induced focal areas of aberrant vessel patterns with increased vessel density and diameter. The stable expression of red-fluorescent protein (td-tomato) by the S24 line allowed dual-color UM to assess the relation of cellular growth patterns and microvessel anatomy (*Figure 2e*).

## Tumor vessels show an increased diameter and density

For the quantification of UM data, vessels were segmented on single image planes and quantified using Matlab (*Figure 3a*). We found a progressive increase in vessel caliber 3 weeks post-tumor implantation compared to healthy regions (mean vessel diameter in outside regions: 3.2 μm; tumor region week 2: 11.5 μm, week 3: 14.7 μm; p<0.001, *Figure 3b*). Also, the tortuosity of tumor vessels was higher (tortuosity in outside regions: 1.11; tumor: 1.16; p=0.05; *Figure 3c*). Vessel density showed a trend toward higher densities within the tumor (0.16 *vs.* 0.11, n.s. *Figure 3d*). The assessment of vessel parameters in entire 3D data sets (100–200 images/animal) was performed with Amira software and showed likewise a significant increase of vessel radii and vessel pathlength within the tumor area compared to outside regions (*Figure 3e–h*). Large, CD31 positive microvessels with increased tortuosity were also confirmed within the tumor by immunohistochemistry (*Figure 3—figure supplement 1a,b*). In the tumor, however, some CD31-positive vessels were negative for lectin-FITC, indicating non-perfused vessels, whereas outside of the tumor all microvessels were double positive for CD31 and lectin-FITC (*Figure 3—figure supplement 1c*).

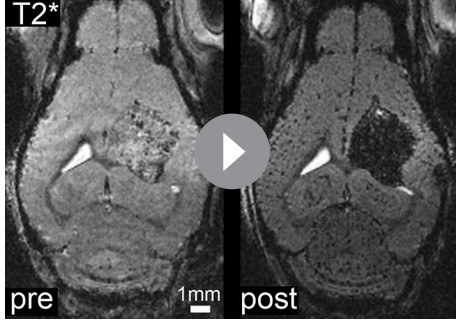

**Video 1.** T2*-w image stack pre- and post-Gd contrast from a mouse 3 weeks after GL261 tumor implantation.

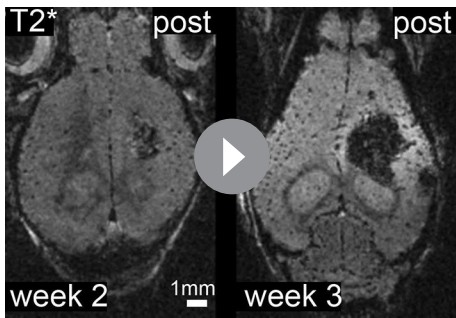

**Video 2.** T2*-w post-contrast image stack 2 and 3 weeks after GL261 tumor implantation.

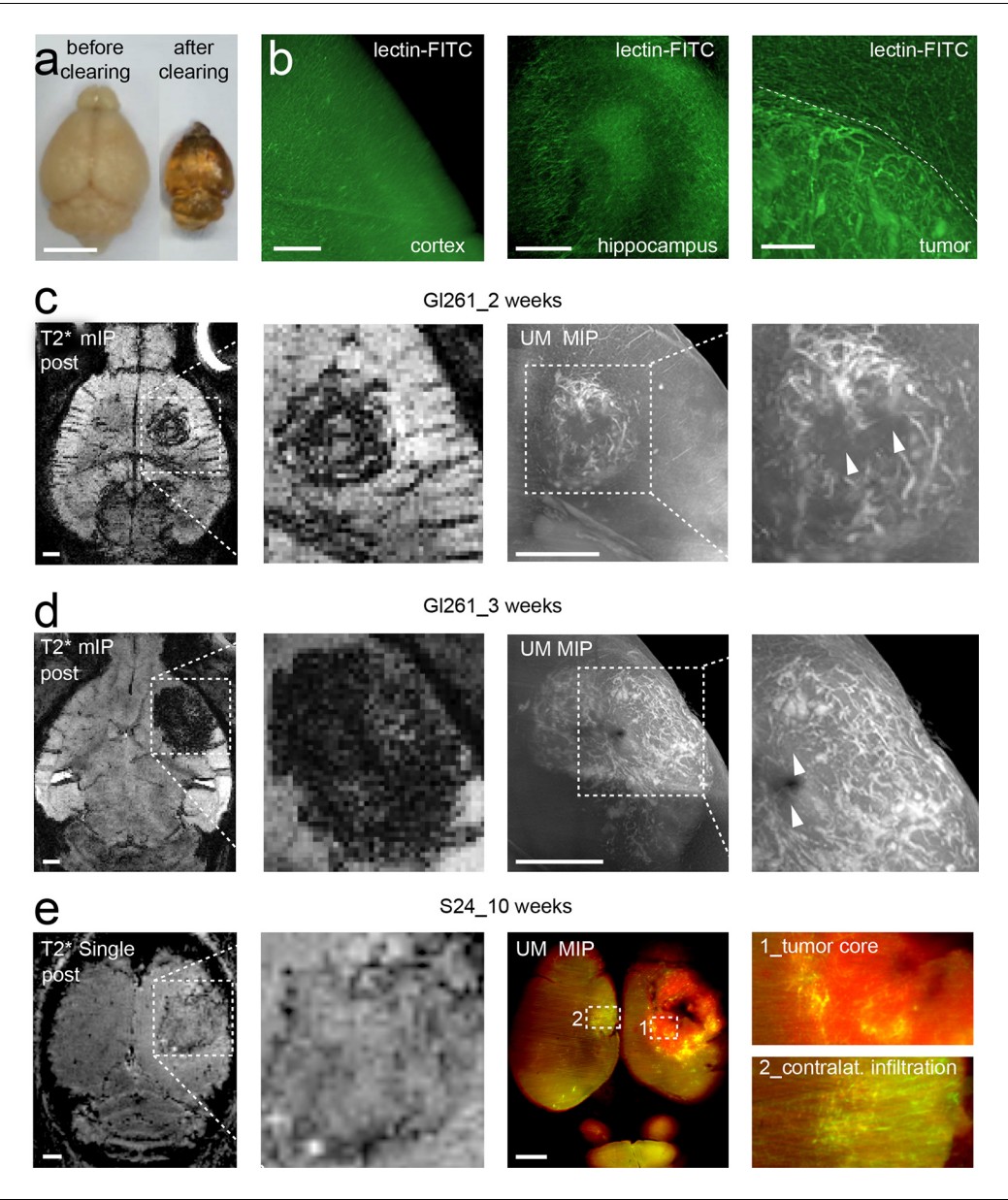

**Figure 2.** Correlated MR-UM provides complementary information of the tumor vascular architecture. Illustration of the mouse brain before and after clearing using the 3DISCO protocol. The brain shrinks by ~40% in size during the clearing protocol (**a**). Cleared UM images of lectin-FITC stained microvessels. Images show the healthy cortex (left) and hippocampus (middle). The glioma-stroma border (dotted line) is depicted on the right image (**b**). T2*-w images and correlative UM images 2 weeks, (**c**) and 3 weeks (**d**) after GL261 tumor implantation. Arrowheads indicate areas of necrosis. T2*-w image and correlative UM images 10 weeks after S24:td-tomato implantation. Inner necrotic tumor areas around the injection track lack fluorescent signal (**e**). The microvasculature is stained with lectin-FITC. MIP: maximum intensity projection, mIP: minimum intensity projection. Scale bar in (**a**) is 5 mm and 1 mm for MR images. For UM scale bars are 250 μm in (**b**) and 1 mm in (c-e).

The following figure supplements are available for figure 2:

**Figure supplement 1.** Lectin-FITC and lectin-texas red staining in healthy mice.

**Figure supplement 2.** Correlated permeability imaging using MR-UM.

**Figure supplement 3.** Time course of S24 tumor development.

## Using correlated T2*-w and UM to quantify antiangiogenic treatment responses

To investigate the power of MR-UM for treatment monitoring, we inhibited the vascular endothelial growth factor with a murine antibody and monitored treatment effects on the vascular compartment. We found that neovessel-formation was partially blocked by VEGF inhibition (*Figure 4a,b* and *Video 6*). Also, the vascularized area and BBB-D decreased as assessed by T2*-w and DCE imaging (*Figure 4c–f*). UM quantification confirmed a partial block of pathological vessel features (vessel diameter and density) by VEGF inhibition (mean vessel diameter under VEGF treatment: 8.3 µm; isotype control: 13.2 µm; p<0.001; *Figure 4g,h*). The beneficial effect on the vascular compartment was specific for VEGF targeted therapy, as cytotoxic treatment with photon irradiation did not lead to signs of vascular normalization (*Figure 4—figure supplement 1*). Despite the efficient blockage of neoangiogenesis, anti-VEGF therapy did not lead to reduced tumor growth (tumor size 1 week after VEGF inhibition: 0.22 mm$^2$ vs 0.22 mm$^2$ in the isotype control group; p>0.05; four mice per group), reflecting current clinical data (*Chinot et al., 2014*; *Taal et al., 2014*).

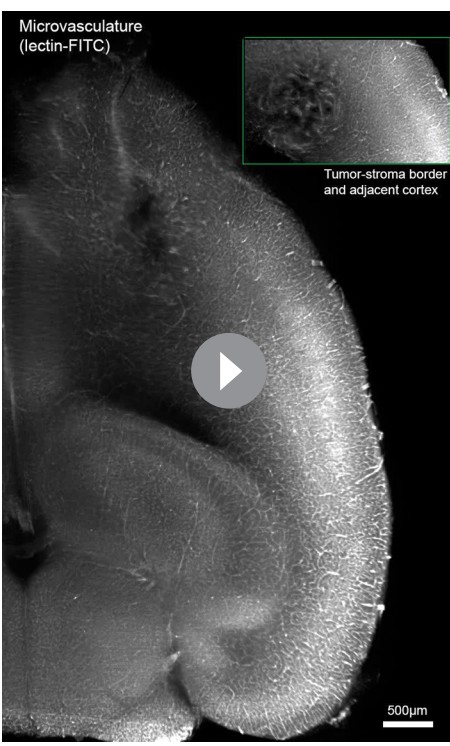

**Video 3.** Ultramicroscopy movie from a mouse injected with lectin-FITC and cleared using the FluoClearBABB protocol. Magnified image shows the tumor-stroma border and adjacent cortex (green box) from a mouse 2 weeks after GL261 tumor implantation.

## Discussion

In this study, we combined in vivo 9.4 Tesla MRI with ex vivo UM and tissue clearing to dynamically resolve tumor arterioles and venules as well as the capillary bed of mouse gliomas at high resolution. Contrast-enhanced T2*-w imaging allowed the visualization of neovessels in the 50–100 µm range. In the GL261 model, angiogenesis started 2 weeks post-tumor implantation and massively increased until week 3. One limitation of MRI concerns its resolution, which was limited to ~80 µm in our study. To overcome this limitation, we employed correlated 3D UM of cleared specimen. For UM, we applied iv perfusion with fluorescent lectins that

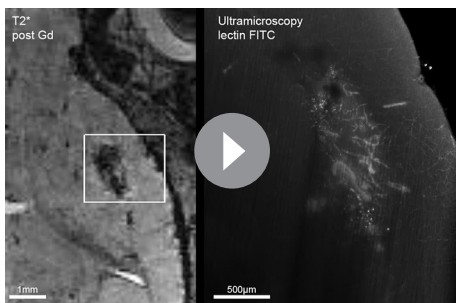

**Video 4.** Correlated MR and UM stack 3 weeks after GL261 tumor implantation. Areas of necrosis are visible on MR and UM and are devoid of vessels. Box indicates the region of UM. UM data is shown as maximum intensity projection of six single images.

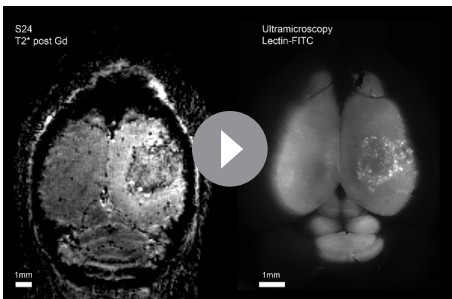

**Video 5.** Correlated MR and UM stack 10 weeks after S24 tumor implantation. UM data is shown as maximum intensity projection of 10 single images.

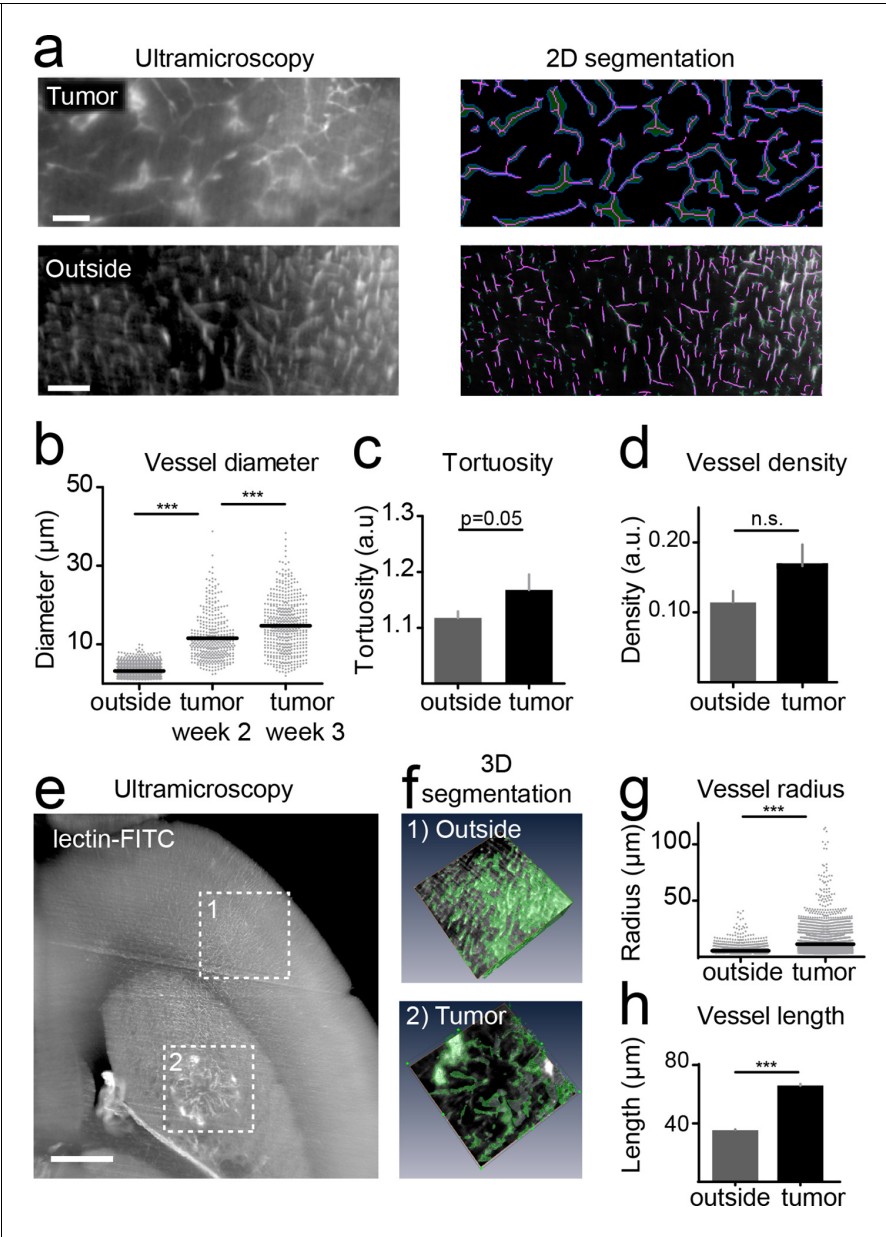

**Figure 3.** Quantification of neoangiogenesis by ultramicroscopy. Representative single plane image of a cleared brain in the tumor and an 'outside' region. Illustration of the vessel segmentation using the 'tubeness' plugin. Vessel segments (magenta) and vessel outline (green) are used to determine the vascular diameter (a). Quantification of the vessel diameter, tortuosity and vessel density (b–d). Illustration of vessel segmentation in 3D (e,f). Vessel radii (g) and pathlength (h) are shown. Scale bars = 100 µm in (a) and 500 µm in (e).

The following figure supplement is available for figure 3:

**Figure supplement 1.** Histological assessment.

bind specifically to endothelial glycoproteins (*Wälchli et al., 2015*) and subsequent optical clearing of whole adult mouse brains using the well-established 3DISCO and the recently published Fluo-ClearBABB protocol (*Ertürk et al., 2012*; *Schwarz et al., 2015*). We obtained high-quality images with both clearing methods that were used to reconstruct the entire tumor anatomy and blood vessel architecture including bulk mass and invasive zone. Tissue clearing and UM has not been employed for the study of neurooncological disease. Previous UM studies investigated microvessels under physiological conditions (*Jährling et al., 2009*) and in the context of solid visceral tumors

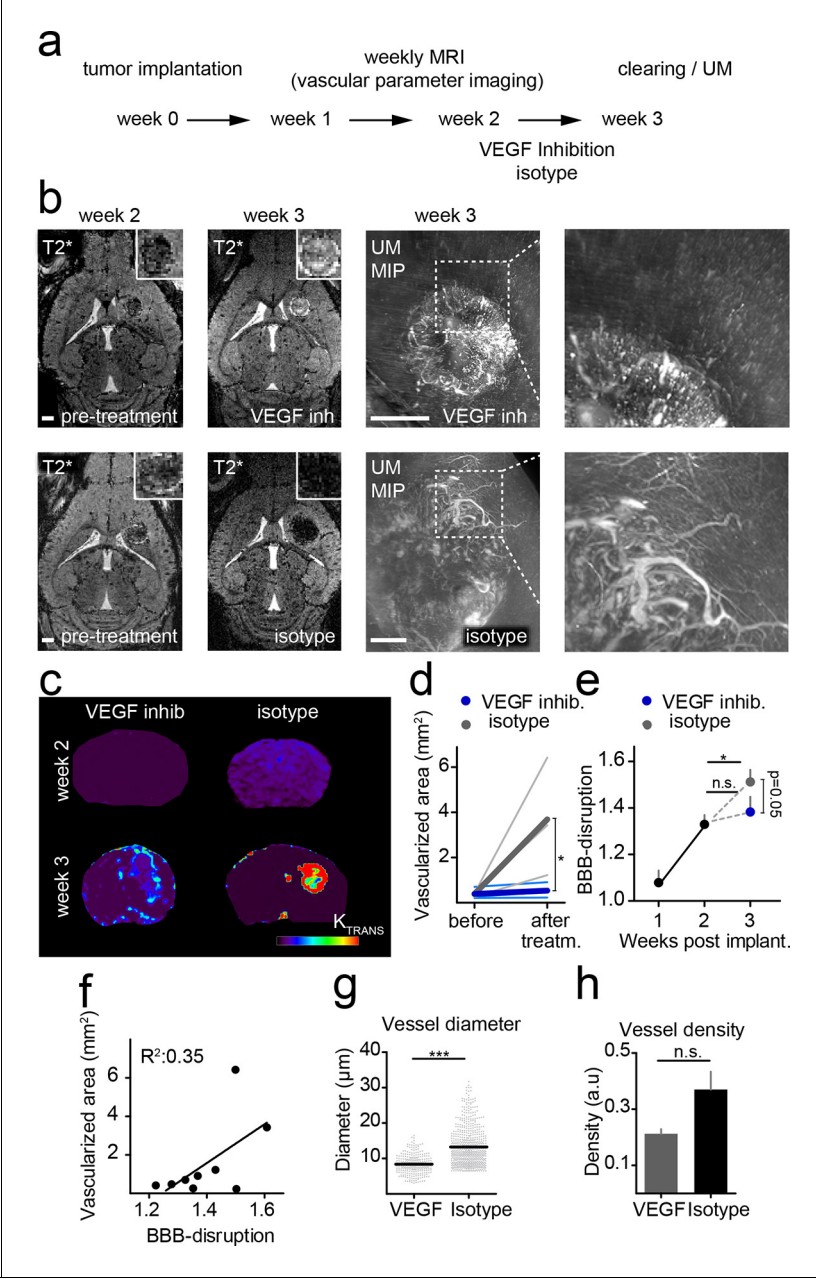

**Figure 4.** Monitoring treatment effects of VEGF inhibition on glioma vessels using MR-UM. Experimental outline (**a**). Single plane, T2*-w images before (week 2) and after VEGF or isotype control treatment (week 3). Treatment was initiated 2 weeks after tumor implantation when a solid tumor component had formed as confirmed on MRI. Correlative UM is shown of the same animal (**b**). Permeability ($K_{trans}$) maps, calculated from DCE MRI are depicted in (**c**). Quantification of the vascularized area on T2*-w images (**d**). Quantification of the blood-brain barrier disruption (BBB-D) on DCE images (**e**). Correlation of BBB-D and the vascularized area (**f**). Quantification of vessel diameter and vessel density on UM images (**g,h**). MIP: maximum intensity projection. Scale bars are 1 mm on MR images and 500 µm on UM images.

The following figure supplement is available for figure 4:

**Figure supplement 1.** Irradiation does not reduce tumor vascularization.

(**Dobosz et al., 2014**). We have extended this work to the field of neurooncology and combined it with advanced MRI techniques. This bridges the gap between MRI and whole brain optical microscopy, which represent separated domains so far. Optical clearing methods have recently undergone a rapid development and evoked large interest in the neuroscience community as they promise to

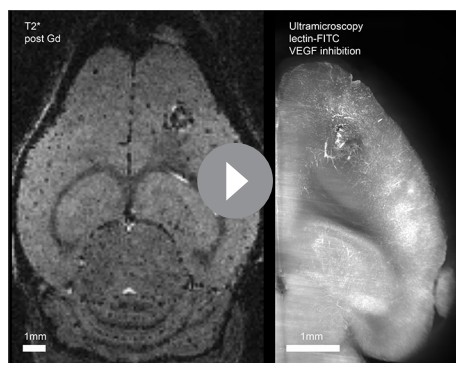

**Video 6.** Correlated MR and UM stack 3 weeks after tumor implantation. Animal was treated with a mouse VEGF inhibitor every 3 days (10 mg / kg) starting 2 weeks after tumor implantation.

allow 'connectomic' and circuit reconstruction studies (*Chung et al., 2013*; *Dodt et al., 2007*; *Ertürk et al., 2011*; *Keller and Dodt, 2012*; *Schwarz et al., 2015*; *Susaki et al., 2014*). Clearing techniques are, however, also amenable to the study of pathological states: First reports have highlighted how 3D information obtained by tissue clearing can give new insights into pathophysiology of neuroinflammatory and neurodegenerative disease (*Jährling et al., 2015*; *Spence et al., 2014*).

The combination with MR, the most widely used clinical imaging technique, offers added value by correlating spatiotemporal information obtained by MR with high-resolution optical imaging. In the preclinical arena MR and UM with tissue clearing appear as natural partners as both produce 3D datasets, which can be compared side-by-side. The advantages of both techniques can be combined: Namely, MR is a versatile tool for longitudinal in vivo studies. UM of cleared specimen can resolve cellular and subcellular processes and utilize the vast toolbox of genetic and chemical fluorescent labels with their high molecular specificity, thus complementing the information gained by MRI. Also, a combination of the two techniques allows cross-correlation of in vivo signals with molecularly defined optical methods. To show the additive power of both methods, we used a neovascularization paradigm since neovessels are hallmarks of many diseases including cancer (*Carmeliet and Jain, 2011*; *Goveia et al., 2014*). Using cross-correlated MR-UM we achieved single vessel resolution and found pathological vessel permeability, tortuosity, and calibers, all of which could be partially reversed by VEGF treatment.

Our data indicate that similar to the human situation, mono-modality antiangiogenic treatment is not sufficient to halt glioma growth. Thus, novel antiangiogenic treatment regimes, possibly in combination with chemotherapy in recurrent disease (as done with promising results in the Dutch Belob Trial (*Taal et al., 2014*), and the ongoing EORTC 26,101 and 26,091 trials), or in combination with additional treatment modalities (i.e.immunotherapy, chemotherapy) are warranted (*Chinot et al., 2014*; *Kamoun et al., 2009*). Anti-angiogenic treatment development and clinical trials require imaging biomarkers to assess vascularization status and treatment effects. The developed T2*-w sequence is suitable to map tumor arterioles and venules in a preclinical setting. Future studies at high-field clinical MR systems should address a possible translation of our MRI approach to the clinical arena.

One limitation of our study relates to the differentiation of the various hierarchical parts of the vascular system. Tumors harbor a manifold of different vascular compartments (veins and arteries of different calibers and sizes with highly variable, but mostly pathological perfusion). Neither MR nor optical methods are able to capture the entire complexity of these compartments. Also, we found that in the bulk tumor some CD31 positive vessels were negative for lectin-FITC, indicating that some tumor vessels were hypo- or non-perfused. This implies that the visualization and quantification of the microvasculature by lectin-injection might slightly underestimate the vascular tumor compartment. The fact that hypo- or non-perfused vessels with abnormal flow exist in tumors is well established (*Carmeliet and Jain, 2011*). Another drawback might be that clearing methods may lead to an alteration in brain size, that is, due to dehydration. For 3DISCO, we quantified the apparent reduction of brain volume using CT imaging before and after clearing. This revealed a volume reduction of ~40%, which however occurred in a uniform fashion and did not change macroscopic tissue proportions. Anatomical structures and substructures of the brain remained fully intact. Vessel diameters obtained by UM could therefore be corrected by a multiplication factor of ~1.6. However, microscopic shrinkage may slightly differ from the macroscopic factor obtained by CT.

Also, when using endogenous fluorescent proteins such as GFP or RFP in combination with exogenous fluorescent markers such as FITC, preservation of fluorescent signals during the clearing process might vary and should be carefully controlled for.

In summary, the present study shows the additive power of correlative MR and UM for assessing the vascular system and the dynamic changes that occur in tumors over time. The applied principles should be easily transferable to additional targets like the immune cell compartment that are amenable to MR and optical labeling strategies (*Kircher et al., 2003*; *Weissleder et al., 2014*). Furthermore, MR-UM should be useful for translational approaches to aid preclinical therapy development. In our study, mono-antiangiogenic treatment with a murine VEGF inhibitor was insufficient to halt tumor growth which mirrors current human studies (*Chinot et al., 2014*). Dual angiogenesis inhibitors are currently being developed and therapeutic effects could be assessed in detail using our toolkit. Thus, we believe that MR-UM can provide a versatile platform for translational research approaches to cross-correlate MR and optical signals.

## Acknowledgements

We are grateful for the kind gift of the murine VEGF inhibitor (Roche, pRED Innovation Center, Penzberg) and of USPIOs (Ralph Weissleder, Massachusetts General Hospital, Harvard Medical School). We thank Martin Schwab (University of Zurich) for critical discussion of the manuscript. We acknowledge support from the DKFZ Light Microscopy Core Facility. MOB, FTK and AH were supported by a physician-scientist fellowship of the Medical Faculty, University of Heidelberg. MOB and FTK are further supported by the Hoffmann-Klose Foundation (University of Heidelberg). MOB acknowledges funding by Neurowind e.V. This study was further supported by funds of the Chica and Heinz Schaller (CHS) Foundation (BT).

## Additional information

### Funding

| Funder | Author |
| --- | --- |
| Medical Faculty, University of Heidelberg | Michael O Breckwoldt<br>Felix T Kurz<br>Angelika Hoffmann |
| CHS Stiftung | Björn Tews<br>Julia Bode |

The funders had no role in study design, data collection and interpretation, or the decision to submit the work for publication.

### Author contributions

MOB, JB, FTK, KO, TK, FW, MP, MB, BT, Conception and design, Acquisition of data, Analysis and interpretation of data, Drafting or revising the article; AH, GS, Acquisition of data, Drafting or revising the article, Contributed unpublished essential data or reagents; MO, Conception and design, Acquisition of data, Analysis and interpretation of data; KD, KK, SC, AA, SH, Acquisition of data, Analysis and interpretation of data, Drafting or revising the article; DS, Conception and design, Acquisition of data, Drafting or revising the article; MF, XH, DM, Acquisition of data, Analysis and interpretation of data, Drafting or revising the article, Contributed unpublished essential data or reagents; WW, Conception and design, Analysis and interpretation of data, Drafting or revising the article

### Ethics

Animal experimentation: All animal experiments were approved by the regional animal welfare committee (permit number: G187/10, G188/12 and G145/10, Regierungspräsidium Karlsruhe).

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
