## [Decision Letter]

Thank you for submitting your work entitled "Correlated MR imaging and ultramicroscopy (MR-UM) is a tool kit to assess the dynamics of glioma angiogenesis" for consideration by *eLife*. Your article has been reviewed by three peer reviewers, and the evaluation has been overseen by a Reviewing Editor and Senior Editor.

The reviewers have discussed the reviews with one another and the Reviewing editor has drafted this decision to help you prepare a revised submission.

The following individuals involved in review of your submission have agreed to reveal their identity: Pascal Sati and Hans-Ulrich Dodt (peer reviewers). A further reviewer remains anonymous.

Summary:

Breckwoldt et al. investigate angiogenesis in a mouse glioma model. They clearly show that the combination of MRI and ultramicroscopy can provide both longtime observation and high resolution of the tumor as well as the effect of VEGF treatment and photon irradiation.

Essential revisions:

1) Methodology: a major concern is the use of dye injection to assess tumor vascularization. The use of contrast agents is of course justifiable for MR, but to correlate in vivo data with ex vivo structural quantifications, authors should rule out the possible involvement of perfusion in the diffusion of the injected dyes. For a thorough anatomical correlation in the same samples, authors should use well-characterized protocols of whole-mount immunohistochemistry (IHC) using specific vascular markers (e.g. CD31, such as used by iDISCO), which would ascertain that the whole vasculature is visualized. The use of IHC would also allow better staining of smaller capillary beds in which perfusion might have been affected by the presence of the tumor. One option would be to validate, in a subset of samples, the structural data with anti-CD31 IHC. Finally, vascular permeability should be assessed/validated using tracers and conventional methods (e.g., conjugated dextrans, Evans blue) such that, as for tumor angiogenesis and vascular morphology, authors could correlate in vivo and ex vivo data.

2) Interpretations: In the manuscript, one can read terms such as "high-resolution quantification", "single vessel resolution", "ultramicroscopy", or "visualization of venules". However, according to MR data, such resolution is not reached. Larger vessels can indeed be observed at the brain surface under MR, but the resolution is pretty poor (e.g., magnifications in Figure 1, Figure 1, and Figure 2). Moreover, from panels in Figure 1, one can ask whether the strong signals are due to presence of vessels or to tracer leakage. In addition, it is dangerous to talk about "neovessels" and "massive angiogenesis" from MR data in Figure 1. Knowing that this MR intrinsically lacks resolution, the reader is hardly convinced that these signals are actually blood vessels. The fact MR relies on perfusion, it seems difficult to decipher between newly formed vessels or increased perfusion in the tumor area. It is only from Figure 2, in correlative experiments with cleared brains, that the reader starts to observe increased vascularization in tumor. Moreover, strong terms such as "increased caliber and tortuousness" are used to describe Figure 1 but none of these features appear in the panels. Finally, it is puzzling why authors chose to analyze "single image planes" from light sheet microscopy in order to quantify vascular morphology. The power of this imaging technique should be leveraged to perform 3D analysis and obtain more detailed measures within tissue volumes (effects would probably reach significance if analyzed in 3D).

3) Translational aspect: Anti-angiogenic therapy to target tumor growth is a classic approach and there is a need to develop better tools to investigate tumor vascularization. However, since apparently the anti-VEGF treatment did not reduce tumor size (as claimed in the text, but unfortunately not illustrated), the rationale for this translational approach is unclear. The authors also mention: "Despite the efficient blockade of neoangiogenesis, anti-VEGF therapy did not lead to reduced tumor growth (…), reflecting current clinical data". This is very vague and one can wonder where these data come from (Method? Number of animals? Graphs? Ref?). Maybe authors can offer an interpretation of why it is failed using their vessel imaging data?

4) An additional technical detail we would mention is that the problem of fluorescence preservation is only severe for endogenous fluorescent proteins like GFP, not really for exogenous fluorescent markers that are much more stable.

---

## [Author Response]

*1) Methodology: a major concern is the use of dye injection to assess tumor vascularization. The use of contrast agents is of course justifiable for MR, but to correlate in vivo data with ex vivo structural quantifications, authors should rule out the possible involvement of perfusion in the diffusion of the injected dyes. For a thorough anatomical correlation in the same samples, authors should use well-characterized protocols of whole-mount immunohistochemistry (IHC) using specific vascular markers (e.g. CD31, such as used by iDISCO), which would ascertain that the whole vasculature is visualized. The use of IHC would also allow better staining of smaller capillary beds in which perfusion might have been affected by the presence of the tumor. One option would be to validate, in a subset of samples, the structural data with anti-CD31 IHC.*

As suggested by the reviewers we have performed immunohistochemistry for CD31 in tissue sections from lectin-injected animals. This resulted in a perfect co-localization of both stainings outside of the tumor mass. In the tumor, however, there were a number of vessels that stained only positive for CD31 but had no apparent lectin staining, indicating non-perfused vessels that were not reached by the lectin injection. We calculated that 49.3% of vessels were double positive for lectin and CD31 within the tumor (179/363 tumor vessels, n=3 mice). 50.7% were only positive for CD31 but not for lectin. The fact that hypo- or non-perfused vessels with abnormal flow exist in tumors is well established (Carmeliet and Jain, 2011). Our utilized method using injected lectins assesses perfused blood vessels and does not account for all tumor vessels. We have added this information to the Results and Discussion and include immunohistochemistry data in the new Figure 3–figure supplement c.

*Finally, vascular permeability should be assessed/validated using tracers and conventional methods (e.g., conjugated dextrans, Evans blue) such that, as for tumor angiogenesis and vascular morphology, authors could correlate in vivo and ex vivo data.*

We have performed correlated in vivo and ex vivo imaging to assess permeability using Gd and Evans blue (EB) (n=2 mice). We find a good agreement of both tracers that show the increased permeability in the tumor area (correlation coefficient: 0.87, p<0.001). Example images are shown in the new Figure 2—figure supplement 2. One concern would be certainly if there was major leakage of lectin into the extracellular space due to blood-brain-barrier disruption (BBB-D). This however, we don’t see in our model, probably because lectins are much bigger than either gadodiamide or EB and do not passively cross the compromised BBB.

*2) Interpretations: In the manuscript, one can read terms such as "high-resolution quantification", "single vessel resolution", "ultramicroscopy", or "visualization of venules". However, according to MR data, such resolution is not reached. Larger vessels can indeed be observed at the brain surface under MR, but the resolution is pretty poor (e.g., magnifications in Figure 1, Figure 1, and Figure 2). Moreover, from panels in Figure 1, one can ask whether the strong signals are due to presence of vessels or to tracer leakage. In addition, it is dangerous to talk about "neovessels" and "massive angiogenesis" from MR data in Figure 1. Knowing that this MR intrinsically lacks resolution, the reader is hardly convinced that these signals are actually blood vessels. The fact MR relies on perfusion, it seems difficult to decipher between newly formed vessels or increased perfusion in the tumor area. It is only from Figure 2, in correlative experiments with cleared brains, that the reader starts to observe increased vascularization in tumor. Moreover, strong terms such as "increased caliber and tortuousness" are used to describe Figure 1 but none of these features appear in the panels.*

We agree with the reviewers that MRI has intrinsic limitations in terms of resolution (like any other imaging technique). The judgment to call the hypointense structures in Figure 1 vessels is based on the fact that most of these hypointense structures are only seen after intravenous contrast administration. We now provide subtraction images of pre and post T2* scans of tumor animals to show the vessel outline that occurs after contrast administration (Figure 5). The concept of blood pool imaging by MR contrast agents is not new (Dosa et al., 2011) but has not been widely exploited in the context of tumor vasculature imaging. We agree with the reviewer that MR images need to be interpreted with caution as susceptibility signals are not specific to the vasculature but can be also caused e.g. by bleeding or calcification. Therefore, careful controls and pre- and post-contrast imaging are necessary for interpretation. In our opinion, however, the term “neovessels” and “increased vascularity” seems justified also based on MRI. Our careful longitudinal study allows us to denominate newly appearing hypointensities as neovessels because such structures only appear after contrast administration and can be blocked by VEGF inhibition (longitudinal imaging of one animal, Figure 5).

Author response image 1.MR-UM of tumor vascularization. a) T2* subtraction images of an animal two and three weeks after Gl261 tumor implantation. b) Occurrence of vessel-like T2* signals over time. c) Tumor size before (week two) and after VEGF or isotype control treatment (week three).**DOI:**
http://dx.doi.org/10.7554/eLife.11712.021

The term “neoangiogenesis” is mostly referring to the microvasculature (5-10 µm vessels) which we cannot visualize directly by MRI. The vascular signals on MRI most likely represent dilated tumor arterioles and venules that are a result of neoangiogenesis but are not strictly neoangiogenesis itself. We have therefore eliminated the term “neoangiogenesis” from the MRI result section. Permeability and leakage of contrast material into the tissue is certainly a concern. We believe, however, that the linear, vascular-like patterns of the post contrast images (e.g. Figure 1) show the tumor blood pool and not mere leakage (as this should result in a uniform signal decrease within the entire tumor). Nevertheless, we have taken care to not “over-interpret” our MR data and have eliminated interpretations from the Result section. We now refer to vascular patterns mostly in the ultramicroscopy section and the Discussion.

*Finally, it is puzzling why authors chose to analyze "single image planes" from light sheet microscopy in order to quantify vascular morphology. The power of this imaging technique should be leveraged to perform 3D analysis and obtain more detailed measures within tissue volumes (effects would probably reach significance if analyzed in 3D).*

We now perform 3D analysis of entire ultramicroscopy data stacks (100-200 single planes) using Amira software. The vasculature is segmented in 3D and vessel radii and pathlength are quantified. Both parameters show highly significant differences between tumor and healthy brain. This data is now included in the extended Figure 3 and described in the methods and Results section.

*3) Translational aspect: Anti-angiogenic therapy to target tumor growth is a classic approach and there is a need to develop better tools to investigate tumor vascularization. However, since apparently the anti-VEGF treatment did not reduce tumor size (as claimed in the text, but unfortunately not illustrated), the rationale for this translational approach is unclear. The authors also mention: "Despite the efficient blockade of neoangiogenesis, anti-VEGF therapy did not lead to reduced tumor growth (*…*), reflecting current clinical data". This is very vague and one can wonder where these data come from (Method? Number of animals? Graphs? Ref?). Maybe authors can offer an interpretation of why it is failed using their vessel imaging data?*

As described in Figure 4 we performed VEGF or isotype control treatment, monitored by MRI (4 mice per group, i.p. treatment with 10 mg/kg VEGF inhibitor or IgG control, every three days). The tumor size increased significantly after one week of treatment (p=0.02) and was not significantly reduced by VEGF treatment compared to the isotype control group (mean tumor size before treatment: 0.076 cm2; after VEGF inhibitor treatment: 0.22 cm2; after isotype treatment: 0.22 cm2; p=n.s., Figure 5). Along these lines two recent clinical trials did *not show* an increased overall survival for bevacizumab treatment compared to standard radio-chemotherapy treatment for primary or recurrent glioblastoma (Chinot et al., 2014; Taal et al., 2014). This information is now included in the Results section. There has been speculation that inhibition of angiogenesis and vascular normalization leads to an activation of tumor stem cells with potential deleterious effects (Lathia et al., 2015). Also our ultramicroscopy data showed only partial vascular normalization and VEGF inhibition did not completely prevent vessel caliber increases and the formation of pathological vessels. This might explain the persistent tumor growth, which was not halted by VEGF inhibition.

*4) An additional technical detail we would mention is that the problem of fluorescence preservation is only severe for endogenous fluorescent proteins like GFP, not really for exogenous fluorescent markers that are much more stable.*

In this manuscript we use td-tomato as fluorescent protein and FITC/Texas-red as fluorescent dyes and both are preserved by our clearing protocols. However we have added the following cautionary note to the Discussion: “When using endogenous fluorescent proteins such as GFP or RFP in combination with exogenous fluorescent markers such as FITC, preservation of fluorescent signals might vary during clearing and should be carefully controlled for.”